# Snow accumulation, albedo and melt patterns following road construction on permafrost, Inuvik–Tuktoyaktuk Highway, Canada

Jennika Hammar[1], Inge Grünberg[1], Steven V. Kokelj[2], Jurjen van der Sluijs[3], and Julia Boike[1,4]

[1]Permafrost Research Section, Alfred Wegener Institute Helmholtz Centre for Polar and Marine Research, Potsdam, Germany
[2]Northwest Territories Geological Survey, Government of Northwest Territories, Yellowknife, NWT, X1A 2L9, Canada
[3]Northwest Territories Centre for Geomatics, Government of Northwest Territories, Yellowknife, NWT, X1A 2L9, Canada
[4]Geography Department, Humboldt-Universität zu Berlin, Berlin, Germany

**Correspondence:** Jennika Hammar (jennika.hammar@awi.de)

**Abstract**

Roads constructed on permafrost can have a significant impact on the surrounding environment, potentially inducing permafrost degradation. These impacts arise from factors such as snow accumulation near the road, which affects the soil's thermal and hydrological regime, and road dust that decreases the snow's albedo, altering the timing of snowmelt. However, our current understanding of the magnitude and the spatial extent of these effects is limited. In this study we addressed this gap by using remote sensing techniques to assess the spatial effect of the Inuvik to Tuktoyaktuk Highway (ITH) in Northwest Territories, Canada, on snow accumulation, snow albedo and snowmelt patterns. With a new, high resolution snow depth raster from airborne laser scanning, we quantified the snow accumulation at road segments in the Trail Valley Creek area using digital elevation model differencing. We found increased snow accumulation up to $36\,\mathrm{m}$ from the road center. The magnitude of this snow accumulation was influenced by the prevailing wind direction and the embankment height. Furthermore, by analysing 43 Sentinel-2 satellite images between February and May 2020, we observed reduced snow albedo values within $500\,\mathrm{m}$ of the road, resulting in a twelve days earlier onset of snowmelt within $100\,\mathrm{m}$ from the road. We examined snowmelt patterns before, during and after the road construction using the normalized difference snow index from Landsat-7 and Landsat-8 imagery. Our analysis revealed that the road affected the snowmelt pattern up to $600\,\mathrm{m}$ from the road, even in areas which appeared undisturbed. In summary, our study improves our understanding of the spatial impact of gravel roads on permafrost due to enhanced snow accumulation, reduced snow albedo and earlier snowmelt. Our study underscores the important contribution that remote sensing can provide to improve our understanding of the effects of infrastructure development on permafrost environments.

## 1 Introduction

Roads constructed on permafrost are at great risk due to the intense climate warming in the Arctic (Rantanen et al., 2022). While climate warming is inducing permafrost thaw at the global scale (Biskaborn et al., 2019), infrastructure may enhance

permafrost degradation at the local scale. Arctic highways alter the local environment, leading to snow accumulation, water ponding at the embankment toe and dust deposition, which are key processes driving permafrost degradation along Arctic highways (Benson et al., 1975; O'Neill and Burn, 2017; Schneider von Deimling et al., 2021). Ground subsidence caused by warming or road effects can have a severe impact on infrastructure leading to increased costs for road maintenance and risk of road failure (Nelson et al., 2001). However, as a result of climate change effects on winter road accessibility (Gädeke et al., 2021) and the increasing regional development in northern regions, there has been a surge in infrastructure construction. For example, half of the more than 200 publicly funded infrastructure projects in the Northwest Territories, Canada, since 2002 are related to roads and highways (Government of Canada, 2023).

Snow drifting in the lee of the road and winter road maintenance lead to snow accumulation at the toe of the embankment (Benson et al., 1975; O'Neill and Burn, 2017). The thermal conductivity of snow, which is low and primarily determined by its density, can range from less than $0.10\,\mathrm{Wm^{-1}K^{-1}}$ for loosely packed, fresh snow to over $0.50\,\mathrm{Wm^{-1}K^{-1}}$ for densely packed and ripened snow (Zhang, 2005). This is five to twenty times lower than the thermal conductivity of mineral soil (Zhang, 2005). Consequently, snow is an effective insulator and prevents the soil from cooling during the winter months (Darrow, 2011; Fortier et al., 2011). However, the insulating effect of snow depends on its depth as well as the timing and duration of snow accumulation. A thin snow cover may cool off the underlying soil because of the high albedo and emissivity of snow (Zhang, 2005) while the insulating effect prevails with increased snow depth (Ge and Gong, 2010), leading to increased soil temperature and thaw depths (Idrees et al., 2015; Park et al., 2015; O'Neill and Burn, 2017). Process-based land surface models showed that variation in ground heat loss, refreezing of the active layer and permafrost thaw are most strongly affected by early season snow accumulation (Park et al., 2015). Furthermore, increased snow accumulation adjacent to northern roads provides an additional source of water when it melts, contributing to higher soil moisture and pond formation along the embankment in the spring. The elevated soil moisture due to impeded drainage and subsidence along embankments, in turn, impacts the subsurface thermal regime by delaying soil refreezing through the release of latent heat (Hinkel et al., 2001; Zhang, 2005).

Arctic highways alter the patterns of snow accumulation, but also affect the snow cover in other ways. In particular, gravel road construction, maintenance and traffic create dust, which settles on the surrounding terrain (e.g., Keller and Lamprecht, 1995; Ackerman and Finlay, 2019; Walker et al., 2022) . The dust deposition is influenced by the wind direction and decreases logarithmically from the road (Everett, 1980; Ackerman and Finlay, 2019; Walker et al., 2022). In winter, road dust reduces the snow albedo, particularly when dust particles accumulate at the snow surface during snowmelt. This results in an earlier spring snowmelt and the earlier onset of soil thaw close to the road (Walker and Everett, 1987).

Due to global warming, the coverage of shrubs has increased and lichen cover has declined in the greater Mackenzie Delta Region of the western Canadian Arctic (Fraser et al., 2014; Nill et al., 2022). This phenomenon can be exacerbated by the presence of roads, as shrubs with relatively low nutrient use efficiencies are favored due to the increased soil nutrient availability resulting from dust deposition and increased soil moisture (Gill et al., 2014; Cameron and Lantz, 2016). Shrubs alter wind speeds and promote snow deposition, altering snow depth and density, and thus the insulating effect of snow cover (Ackerman, 2018). In addition to this potential warming effect, shrub branches also protrude the snow surface in spring, which lowers the albedo and may lead to earlier snowmelt and thus a longer summer season (Marsh et al., 2010; Wilcox et al., 2019). In contrast,

during summer, the presence of shrubs reduces the solar radiation at the soil surface, leading to cooler soils and potentially shallower thaw depths (Lawrence and Swenson, 2011; Myers-Smith and Hik, 2013). However, the reduced availability of light, water and nutrients has a negative impact on mosses growing below the shrubs (Gill et al., 2014). This, together with heavy dust deposition, results in a decline in moss and lichen cover, which may decrease thermal insulation of the ground in summer, further amplifying soil warming (Raynolds et al., 2014).

In the present study, we focus on the Inuvik to Tuktoyaktuk Highway (ITH) in Northwest Territories, Canada (NWT). The highway corridor traverses the ecoclimate gradient of the treeline characterized by a northward decrease in snow and permafrost temperatures (Kokelj et al., 2017) and is characterized by a heterogeneous topography and ice-rich permafrost, which is particularly vulnerable to climate change (Rampton, 1988; Burn and Kokelj, 2009). The recently constructed highway provides a unique opportunity to study the evolution of impacts through time.

In situ measurements, including snow depth or albedo, help to derive important information on processes locally. However, such local information may not be applicable or extendable to another section of the road. At the date of writing, we are not aware of any other studies that have quantified the snow accumulation enhancement along a permafrost gravel road with methods other than point-wise snow depth measurements (Benson et al., 1975; O'Neill and Burn, 2017). However, point observations are typically comprised of a few transects with limited maximum distance to the road. Drone imaging has been used successfully to determine snow depth distribution and snowmelt timing in the Trail Valley Creek (TVC) watershed, also located along the ITH corridor, using structure from motion techniques (Wilcox et al., 2019; Walker et al., 2021). In addition, the use of drone imaging techniques has facilitated the observation and quantification of permafrost processes and infrastructure impacts due to thaw near infrastructure (van der Sluijs et al., 2018). Rapid advances in drone technology have expanded applications beyond line-of-sight (van der Sluijs et al., 2023b) comparable with conventional airborne laser scanner (ALS) coverage. However, to our knowledge, drone-based mapping has not yet been used to assess snow accumulation along Arctic highways and ALS data is typically only collected in summer for infrastructure and thermokarst mapping (van der Sluijs et al., 2018).

Passive satellite microwave-based estimates of snow depth are generally too coarse with spatial resolutions of $25\,\mathrm{km}$ and a saturation around $0.8\,\mathrm{m}$ snow depth, which limits their application where higher spatial resolutions are required (Lievens et al., 2022). Compared to drones, satellite remote sensing offers the advantage of wider spatial coverage and longer time series, enabling the retrieval of physical surface parameters that serve as indicators of permafrost degradation at larger scales. Snow cover percentage can be assessed by optical remote sensing because of its spectral properties, making it distinguishable from snow free areas (e.g., Hall et al., 1995; Macander et al., 2015; Morse and Wolfe, 2015).

Our study utilized remote sensing techniques to investigate the effects of the ITH on snow conditions. We analyzed snow depth, snow albedo and snowmelt patterns using ALS data, Sentinel-2 satellite imagery and Landsat data. The specific objectives of this paper are to (1) quantify the influence of embankment height and aspect on snow accumulation, (2) assess the impact of road dust on snow albedo and its variation with distance from the road and (3) determine how dust has affected snowmelt by comparing the patterns of melt along the road corridor before, during and after ITH construction.

## 2 Datasets & Methods

### 2.1 Study site

Our study site was the ITH, which is a $138\,\text{km}$ all-weather gravel highway to the east of the Mackenzie Delta in NWT, Canada (Fig. 1). The highway's construction began in 2014 and it was officially opened in 2017. The embankment height varies along the road depending on the topography and ice content (Kiggiak - EBA, 2011), reaching heights up to $12\,\text{m}$ (De Guzman et al., 2021). To protect the underlying thaw-sensitive permafrost, the minimum design height was $1.4\,\text{m}$ (Kiggiak - EBA, 2011). This was designed to accommodate consolidation of the initially frozen embankment with the intent of maintaining an active layer above the native permafrost. The embankment includes the $8\,\text{m}$ to $9\,\text{m}$ wide highway and slopes of $33\,\%$ on both sides. Moreover, the highway was built using fill but no cuts into the terrain to minimize the disturbance of the permafrost (Kiggiak - EBA, 2011). Since its official opening, ongoing maintenance efforts have been undertaken on the road to address ground subsidence and embankment settlement. These maintenance activities have involved the replacement and installation of culverts, as well as regular grading to ensure a driveable surface.

The ITH is located within the continuous permafrost zone on the uplands east of the Mackenzie Delta and traverses the treeline zone through the physiographic regions of the Anderson Plain in the south and the Tuktoyaktuk Coastlands in the north (Mackay, 1963; Rampton, 1988). The term "treeline zone" is utilized to refer to the transition area between boreal forest and tundra, implying that the change is not abrupt, and a gradient of tree densities and heights can be observed within this zone (Antonova et al., 2019). The region south of the treeline zone is characterized by open spruce woodlands and peat plateaus. Tree cover decreases northwards and transforms to tundra with tall shrubs at the southern edge of the treeline zone, while the northern edge is characterized by sedges and dwarf shrubs (Burn and Kokelj, 2009). Sedges, grasses, ericaceous shrubs and lichens dominate the low Arctic tundra north of the treeline zone (Burn and Kokelj, 2009). The landscape is characterized by a lake-rich, hummocky and rolling terrain. The subsurface material predominantly originates from ground moraines (fine-grained stony tills) of the Late Wisconsin glacial episode (Duk-Rodkin and Lemmen, 2000) with some interceptions of alluvial, glaciofluvial and lacustrine deposits. The subsurface material is frequently ice-rich and consequently sensitive to climate change (Burn and Kokelj, 2009) as demonstrated by the spatial distribution and increasing activity of retrogressive thaw slumps in this area (van der Sluijs et al., 2023a). The mean annual air temperature for 1990–2020 was $-7.1\,°\text{C}$ and $-8.9\,°\text{C}$ at Inuvik and Tuktoyaktuk, respectively (Environment and Climate Change Canada, 2021). The mean annual ground temperature in undisturbed terrain ranges from approximately $-1\,°\text{C}$ in the taiga regions near Inuvik to approximately $-6\,°\text{C}$ in low-shrub tundra near Tuktoyaktuk (Kokelj et al., 2017). The permafrost thickness ranges from $100\,\text{m}$ near Inuvik to $500\,\text{m}$ in the northern parts of the study area (Mackay, 1967; Judge et al., 1979; Burn and Kokelj, 2009).

### 2.2 Methods objective 1: Snow accumulation at the embankment

To understand the influence of the road on the snow accumulation, we performed differencing between two elevation datasets, one snow covered digital elevation model (further referred to as snow covered DEM) and one snow free digital terrain model (further referred to as snow free DTM). For this analysis, we assumed that only the snow influences the elevation differences.

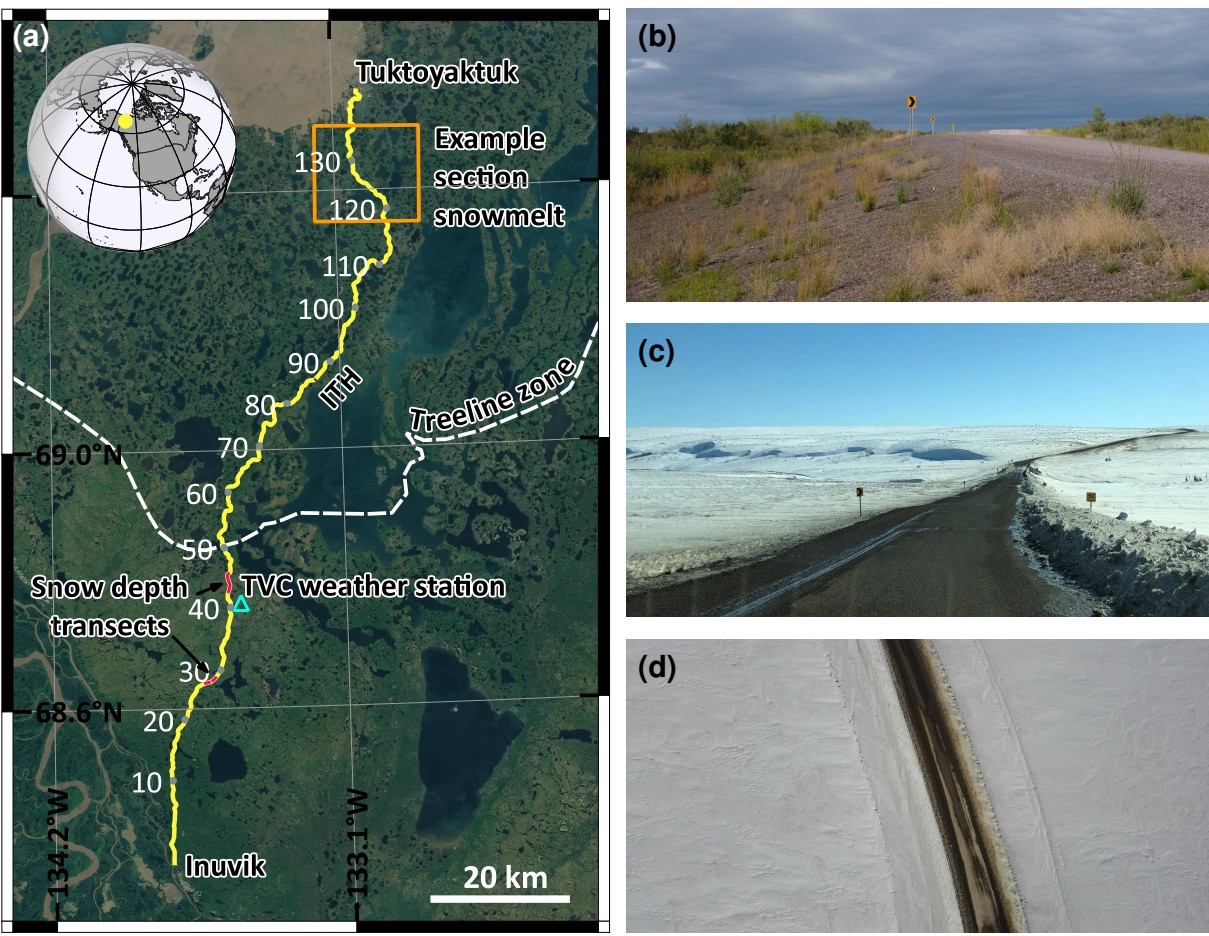

**Figure 1.** (a) The Inuvik–Tuktoyaktuk Highway (ITH) in Northwest Territories, Canada with kilometers shown as white numbers. The location of the snow accumulation analysis close to the TVC research station is highlighted in red (at km 27–28 and 42.5–45.5). The inset map at the top left depicts the location of the study area in North America and the orange square highlights the example chosen for the snow cover image in Fig. 6a,b. The treeline zone (Brandt, 2009) is shown as white dashed line. Base map: Sentinel-2 cloudless data by EOX IT Services GmbH. Photos of the highway on (b) August 24, 2022 (photo credits: I. Grünberg), (c) on April 23, 2022 (photo credits: J. van der Sluijs) and (d) a drone image (km 42) taken with a Sensefly eBee Plus RTK on April 23, 2022 (photo credits: J. van der Sluijs).

We used a snow free DTM by Lange et al. (2021), which was derived from ALS data with average point density of $11.9\,\mathrm{pts/m^2}$
acquired on August 29, 2018 using a Riegl LMS-Q680i onboard the Alfred-Wegener-Institute's (AWI) POLAR 5 research aircraft, a modified Basler BT-67. The coordinate system for the snow free DTM is WGS 1984, UTM 8N (EPSG 32608). The snow free DTM has accuracy and precision levels of $0.03\,\mathrm{m}$ and $0.05\,\mathrm{m}$ respectively, both derived from global navigation satellite system (GNSS) measurements conducted during August 2018 (Lange et al., 2020). However, the snow free DTM likely overestimates terrain elevation in areas of dense vegetation, leading to an underestimation of snow depth in those areas.

The snow covered DEM was derived from a second ALS survey onboard the POLAR-5 aircraft on April 10, 2019, representing the temporal window when the snowpack reaches its maximum depth during spring (Spark, 2023). The laser scanner was a Riegl VQ-580 which is specially designed to measure on snow and ice (Riegl, 2021). To create a DEM from the binary point-cloud data, we extracted the xyz-data and applied an atmospheric backscatter filter using the python-package awi-als-toolbox (Hendricks, 2019). We indexed and chunked the point clouds into tiles with a temporary buffer of $20\,\text{m}$ to avoid edge artifacts

during the classification using LAStools. To distinguish ground points from non-ground points, we applied the Simple Morphological Filter (SMRF) (Pingel et al., 2013) using the Point Data Abstraction Library (PDAL) (PDAL contributors, 2022). We interpolated the ground-classified dataset with inverse distance weighting (IDW) using the Points2Grid approach integrated into PDAL, with the default circular neighborhood search radius of $grid\ resolution \cdot \sqrt{2}$, resulting in a radius of $1.41\,\text{m}$. The format of the final gridded dataset was GeoTIFF with $1\,\text{m}$ cell size and the coordinate system WGS 1984, UTM 8N (EPSG

32608). The method which we have implemented for the ground classification and DEM generation is further described in Bookhagen (2018). In this study, we focused on the intersection of the two DEMs and the ITH spanning $4\,\text{km}$ of the highway (Fig. 1). We subtracted the snow free DTM from the snow covered DEM to obtain the snow depth distribution. The created snow covered DEM and snow depth distribution datasets are published on PANGAEA (Hammar et al., 2023).

   We created $140\,\text{m}$ long transects (n = 4026) perpendicular to the road every $1\,\text{m}$ over the ITH centerline using GRASS

GIS. This ensured comprehensive coverage of the entire section of highway from which we have data. Subsequently, we extracted the snow depth values. Unfortunately, we did not have point measurements of snow depth to validate our dataset. As an alternative approach to assess the accuracy of the snow depth raster, we used the transect sections over the road surface as a reference, assuming that it was free of snow and at the same elevation in both datasets. Consequently, the snow depth should be close to zero over the road. Our snow depth product has a high accuracy with a median snow depth of $<1\,\text{cm}$ at the road

surface and a standard variation of $5\,\text{cm}$ (Fig. 2). Because of the high accuracy of the snow depth product, we did not perform any co-registration or shifting of the snow covered DEM.

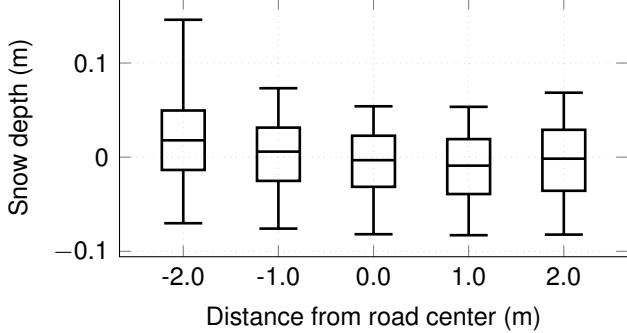

**Figure 2.** Snow depths derived from $5\,\text{m}$ long transects across the road (n = 4026). The center lines indicate the median snow depth of all grid cells at a given distance from the approximate center location of the road, the box covers the 25th to 75th percentile and the whiskers the 5th to 95th percentile.

We assigned each transect an angle between $0\,^\circ$ to $180\,^\circ$ using the field calculator in QGIS. We categorized transects with angles $<45\,^\circ$ or $>135\,^\circ$ to the north–south direction (n = 973), while those with angles within $45\,^\circ$ to $135\,^\circ$ were classified as in line with the east–west direction (n = 3053). In order to determine the embankment height, we required an elevation model without the embankment. For this, we clipped the snow free DTM with a $10\,\mathrm{m}$ buffer around the centerline of the road and interpolated the clipped no data areas with a maximum search radius of $100\,\mathrm{m}$. Next, we subtracted this interpolated no-road DTM from the initial DTM and determined the height difference at the roads centerline for each transect. We divided the embankment heights into equal intervals and categorized them as low (n = 2051, $0.3\,\mathrm{m}$ to $0.9\,\mathrm{m}$), intermediate (n = 1335, $0.9\,\mathrm{m}$ to $1.4\,\mathrm{m}$) and high embankments (n = 640, $1.4\,\mathrm{m}$ to $2\,\mathrm{m}$). We analysed the snow depth distribution based on distance to the road center, cardinal direction and embankment height.

We used ERA5 snow depth data from 1950–2022 (Muñoz Sabater, 2019) with a horizontal resolution of $0.1\,^\circ$ by $0.1\,^\circ$ to assess whether or not April 2019 was representative for typical spring conditions. To achieve this, we calculated the daily average snow depth for each year from the nine nearest pixels surrounding the transects. To our knowledge, there are no ERA5 validation studies of snow depth in the study area and therefore, this assessment was conducted without knowing local accuracy and precision.

## 2.3 Methods objective 2: Road dust and snow albedo

To examine the temporal variation and the spatial extent to which the dust from the ITH affect the snow albedo, we utilized 43 Sentinel-2 images between February 15 and May 30, 2020. We selected Sentinel-2 for this analysis because of the finer spatial resolution in the utilized bands ($10\,\mathrm{m}$ to $20\,\mathrm{m}$) as compared to Landsat ($30\,\mathrm{m}$). Surface snow albedo is affected by surface characteristics such as particle size, water content, impurity content, surface roughness, crystal orientation and structure (Zhang, 2005). Therefore, snow albedo can be strongly reduced by road dust. The broadband albedo is defined as the ratio of the reflected to the incident flux density from a unit surface area for the entire solar radiation spectrum ($300\,\mathrm{nm}$ to $3000\,\mathrm{nm}$) while narrowband albedo is the same ratio at a narrow range of wavelengths (Cogley et al., 2011). To obtain broadband albedo from satellite imagery, four steps are required (Traversa et al., 2021; Ren et al., 2021): (1) Cloud masking and atmospheric correction, which removes the scattering and absorption effects of atmospheric gasses and aerosols, (2) topographic correction, which corrects for the surface slope and aspect effects resulting in reflectance values that would be recorded over the equivalent horizontal surface (Wen et al., 2009), (3) anisotropy correction, which employs bidirectional reflectance distribution functions (BRDF) to correct for the anisotropic reflectance of a surface under varying illumination conditions and (4) Narrowband to Broadband (NTB) conversion.

As we focused on relative differences in albedo and not on absolute values, we did not implement topographic or anisotropic corrections. We used atmospherically corrected Sentinel-2 data (bottom of atmosphere) available in Google Earth Engine (GEE). We pre-processed the Sentinel-2 images using the GEE python application programming interface.

For cloud masking, we used the s2cloudless machine learning-based cloud detector developed by the Sentinel Hub research team, which uses cloud masks created by MAJA (Hagolle et al., 2017) as proxy for ground truth (Zupanc, 2019). Compared to other widely used cloud detection algorithms, s2cloudless has a high cloud detection rate and a lower misclassification

rate of land and snow as clouds (Zupanc, 2021). Cloud probability for each pixel at $10\,\mathrm{m}$ scale is provided for every image in the Sentinel-2 archives in GEE, giving each image a corresponding s2cloudless image. The cloud probability is based on the pixel's Sentinel-2 band values (Zupanc, 2020) and the cloud shadow is defined by cloud projection intersection with low-reflectance near infrared (NIR) pixels (Miceli and Braaten, 2020). In this study, we applied the script by Miceli and Braaten (2020) with $40\,\%$ as threshold for cloud probability, 0.15 as threshold for NIR reflectance, a $50\,\mathrm{m}$ buffer to dilate the edge of cloud-identified objects and a maximum distance of $2\,\mathrm{km}$ to search for cloud shadows from cloud edges.

For the NTB conversion, we used the Liang et al. (2003) albedo formula. While it was initially developed for Landsat 5/7 data, it has been tested with the same empirical weighting parameters for the corresponding bands of Sentinel-2 (Naegeli et al., 2017). The formula provided good estimates for mean albedo values for glaciers (Naegeli et al., 2017) and thus, we expected the same performance for snow covered surfaces. The formula from Liang et al. (2003) adopted to the bands of Sentinel-2 (blue – $b_2$, red – $b_4$, NIR – $b_8$ and the two bands of shortwave infrared (SWIR); SWIR1 – $b_{11}$ and SWIR2 – $b_{12}$) as suggested by Naegeli et al. (2017) is as follows:

$$\alpha_{\mathrm{Liang}} = 0.356\,b_2 + 0.130\,b_4 + 0.373\,b_8 + 0.085\,b_{11} + 0.072\,b_{12} - 0.0018 \tag{1}$$

To obtain the albedo from only the snow, we masked snow free pixels and open water using the normalized difference snow index (NDSI) and normalized difference water index (NDWI), respectively. NDSI helps to discriminate snow, ice and water from bare soils and clouds (Dozier, 1989) by taking advantage of the high reflectance characteristics of snow and ice in the visible spectrum and the absorption in SWIR1. NDSI is defined as the difference between the green ($b_3$) and the SWIR1 ($b_{11}$) reflectance divided by their sum (Hall et al., 1995):

$$NDSI = \frac{b_3 - b_{11}}{b_3 + b_{11}} \tag{2}$$

Following Hall et al. (1995), ice, snow and water were identified by a NDSI threshold value of $>0.4$, whereas bedrock and bare soils were identified by a value of $<0.4$. NDWI was proposed by McFeeters (1996) to delineate open water features. The index aims to use the green band to maximize the reflectance of the water body and the high absorption of NIR wavelengths to minimize it (McFeeters, 1996). It is defined as the difference between the green and NIR bands divided by their sum:

$$NDWI = \frac{b_3 - b_8}{b_3 + b_8} \tag{3}$$

To mask the open water, we applied a NDWI threshold of 0.4, which we determined based on visual interpretation.

For the pixel based analysis, we generated 120 buffers in $5\,\mathrm{m}$ intervals along the vectorized road, excluding the road and embankment. We extracted the snow albedo values in the buffer zones using the extract function in the eo-box Python package (Mack, 2018).

## 2.4 Methods objective 3: Spatial extent of early snow free areas

To determine how road dust affects the spatial extent of snowmelt, we assessed the fraction of snow covered pixels along the corridor of the ITH with Landsat-7 and -8 imagery in May or June, when the landscape was still partly covered by snow. We

decided to use Landsat rather than Sentinel-2 because of the longer time series. We expected that even before road construction, the higher elevated and drier areas, which were chosen for building the ITH, would, on average, have less snow cover than lower lying neighbouring terrain. To isolate the road effect, we compared imagery before (2002-05-17 and 2006-05-19, Landsat-7), during (2015-05-13, 2016-05-06 and 2017-05-18, Landsat-8) and after (2018-05-30, 2019-05-17 and 2020-05-26, Landsat-8) road construction. The imagery for these dates were cloud-free and showed a landscape that was partially covered with snow.

The snowmelt pattern analysis followed the workflow in Fig. 3. First, we calculated the NDSI for each image and created a snow mask. Second, we created a water mask using the NDWI of a snow free image from August 2019 and applied it to all images to remove the water bodies. Using both masks, we divided all pixels into (I) snow free ground, (II) snow and (III) water or ice. Last, we extracted the pixel categories and the pixel coordinates in a $1000\,\mathrm{m}$ buffer zone from the road using the extract function in the eo-box Python package (Mack, 2018). With the pixel coordinates we calculated the distance of every pixel to the road edge using the QGIS tool "distance to nearest hub". To obtain percent snow cover, we divided the pixels that were classified as snow by the total number of land pixel per $5\,\mathrm{m}$ distance along the entire $138\,\mathrm{km}$ of the road.

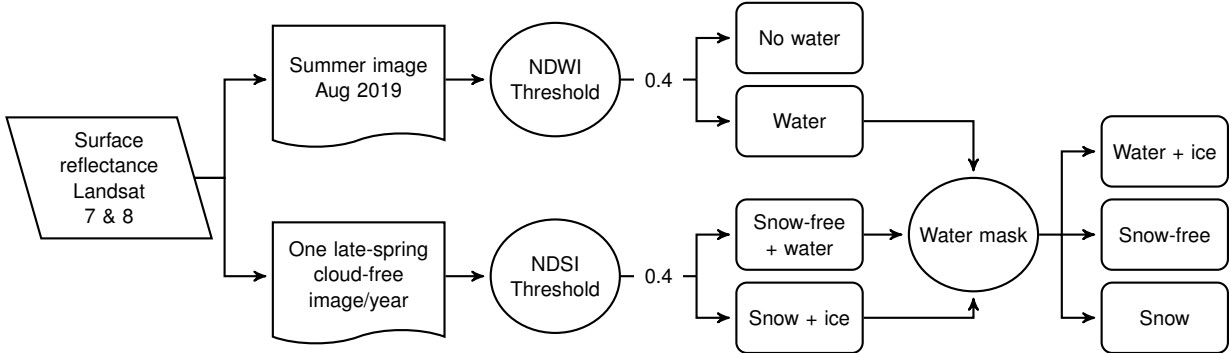

**Figure 3.** Processing chain for snow discrimination using Landsat data.

The total snow cover observed in the Landsat images varies across the different years, due to multiple contributing factors such as photo timing and climate variability, making it difficult to compare the road effect between the years. To focus on the road effect exclusively, we normalized the snow cover by calculating the snow cover average for each year at distances between $900\,\mathrm{m}$ to $1000\,\mathrm{m}$. At this distance, it is reasonable to posit that the impact of the road is negligible (Everett, 1980; Ackerman and Finlay, 2019). We used this average value as a normalizing value and divided each snow cover value by this value.

## 3   Results

### 3.1   Snow accumulation at the embankment

The ITH embankment increased snow accumulation on both sides of the road (Fig. 4a). The region of enhanced snow accumulation reached up to $36\,\mathrm{m}$ from the road center at north-facing embankment slopes with the greatest accumulation at $10\,\mathrm{m}$ to $26\,\mathrm{m}$ (Fig. 4a). Within this $16\,\mathrm{m}$ range, we estimated a median snow depth up to $0.92\,\mathrm{m}$, representing the maximum median

value observed within the $1\,\text{m}$ increments on the transects. In southern direction, we measured enhanced snow accumulation up to $20\,\text{m}$ from the road center and the median snow depth within the highest accumulation zone (between $9\,\text{m}$ to $14\,\text{m}$) was up to $0.88\,\text{m}$ (Fig. 4a). Furthermore, the median snow depth farther away from the road was generally greater on the northern side ($0.5\,\text{m}$) than on the southern side ($0.3\,\text{m}$). As shown by the 25th–75th percentiles, the snow depth variability increased with increasing snow accumulation for both sides. The snow depth variability was generally greater on the northern side than the southern side. Moreover, the northern side showed a double peak in snow depths as represented by the 75th percentiles, whereas the southern side showed only one snow depth peak. The average snow depth at the northern and western sides of the embankment matches well with the regional average snow depth obtained from the ERA5 snow depth product of $0.5\,\text{m}$ at the day of our data acquisition (April 10, 2019). According to the ERA5 dataset, the multi-year average snow depth at this date is $0.6\,\text{m}$ (1950–2022) and thus $0.1\,\text{m}$ more than in April 2019. The ERA5 dataset also showed that the onset of rapid snowmelt did not start until one month later.

As compared to north–south direction, more transects in the study area were located in east–west direction (n=973 versus n=3053). Enhanced snow accumulation reached up to $36\,\text{m}$ and $25\,\text{m}$ from the road center at the western and eastern side, respectively (Fig. 4a). We found the deepest snow within a distance of $10\,\text{m}$ to $20\,\text{m}$ from the road center on the western side (median snow depth up to $1.09\,\text{m}$) and a distance of $10\,\text{m}$ to $16\,\text{m}$ on the eastern side (median snow up to $0.97\,\text{m}$).

The snow accumulation characteristics varied with embankment height (Fig. 4b). The increased snow accumulation for all embankment heights reached distances up to $36\,\text{m}$ (Fig. 4b) from the road center. For low embankments (ranging from $0.3\,\text{m}$ to $0.9\,\text{m}$) the median snow depth was up to $1.03\,\text{m}$ at the embankment toe. For intermediate embankments heights ($0.9\,\text{m}$ to $1.4\,\text{m}$) the median snow depth was up to $1.16\,\text{m}$ and the variation in snow depth was more pronounced at greater distances from the road compared to other embankment heights. The highest embankments, measuring $1.4\,\text{m}$ to $2\,\text{m}$ in height at the studied road segments, exhibited a median snow depth of up to $1.32\,\text{m}$.

At the Trail Valley Creek research station close to the study area, the predominant wind direction between October 2018 and April 2019 was south (Fig. 4c). This period corresponds to the winter months preceding snow depth data acquisition. Eastern and western wind directions occurred approximately equally often, but wind from the east was, on average, stronger. Relative to the typical meteorological conditions recorded between 1998 and 2021, easterly winds were more frequent during the study period (Environment and Climate Change Canada, 2021).

## 3.2 Snow albedo decrease

We found that during the snowmelt period, the snow albedo decreased earlier in areas close to the road as compared to $500\,\text{m}$ away (Fig. 5a,c). Before 5 April, snow albedo was similar at all distances from the road embankment edge ranging from $0.5$ to $0.83$ for different dates. From April 5 to May 8, snow albedo close to the road was already decreasing, while snow albedo at more than $200\,\text{m}$ distance remained high. The largest difference of snow albedo by distance to the road was around May 13, when the closest pixels ($20\,\text{m}$ distance) showed snow albedo values of only $0.45$ while albedo at $900\,\text{m}$ from the road was around $0.7$. Afterwards, the variation gradually decreased again but consistently followed the same pattern, with the closest distance to the road edge displaying the lowest albedo values and the farthest distance showing the highest values.

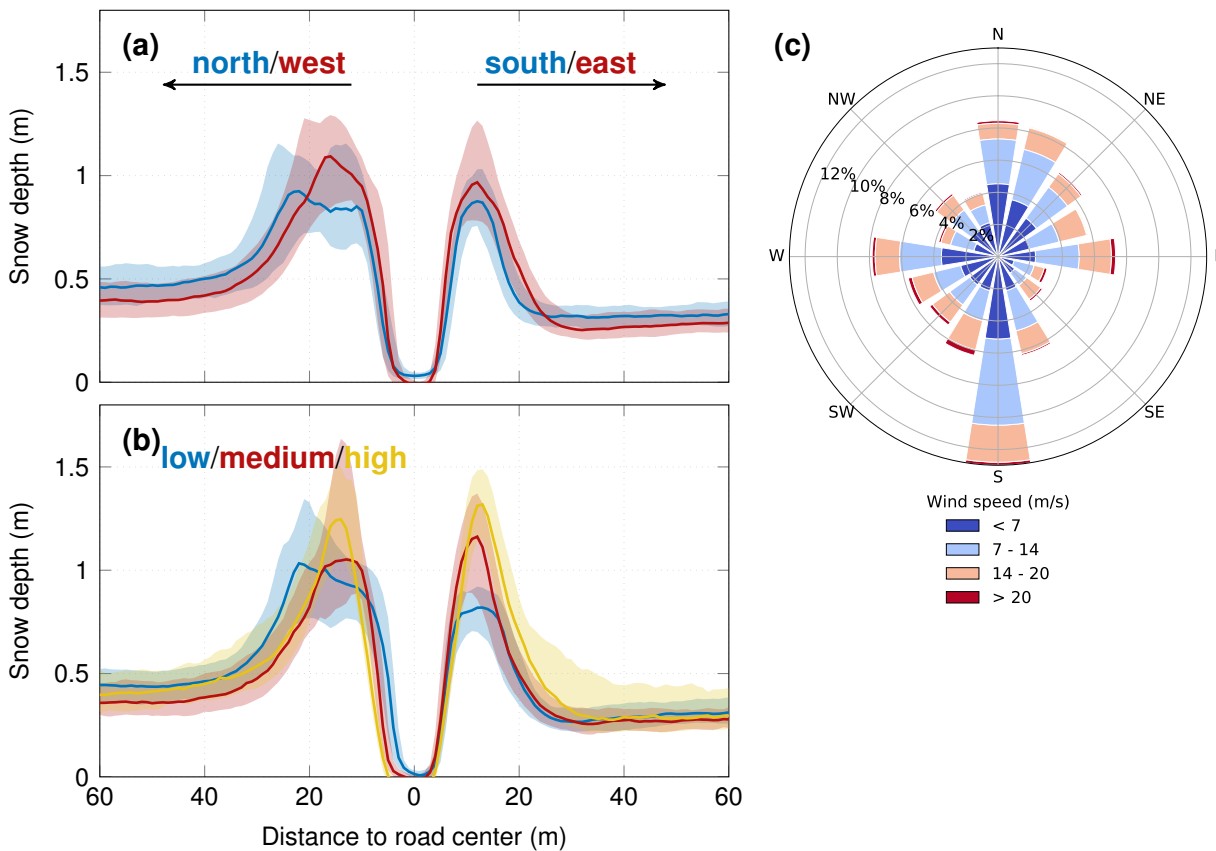

**Figure 4.** Snow accumulation derived from n = 4026 transects across the road at (a) north–south (n = 973, blue) and east–west (n = 3053, red) facing embankment slopes and at (b) low (n = 2051, $0.3\,\mathrm{m}$ to $0.9\,\mathrm{m}$, blue), intermediate (n = 1335, $0.9\,\mathrm{m}$ to $1.4\,\mathrm{m}$, red) and high embankments (n = 640, $1.4\,\mathrm{m}$ to $2\,\mathrm{m}$, yellow). The coloured lines indicate the median snow depth of all grid cells at a given distance from the approximate center location of the road, the shades represent the 25th–75th percentiles. (c) shows the predominant wind direction for the Trail Valley October 2018 – April 2019 (climate identifier: 220N005, current station operator: Environment and Climate Change Canada – Meteorological Service of Canada) (Environment and Climate Change Canada, 2021).

The study area was fully snow covered throughout the winter until May 8 (Fig. 5b, derived from the NDSI). In the following two weeks, the snow cover close to the road decreased drastically. Areas far away from the road edge ($500\,\mathrm{m}$ to $900\,\mathrm{m}$)

remained almost fully snow covered until May 20. Thereafter, the snow melted rapidly and by May 30, undisturbed areas showed a snow coverage of $30\,\%$. Close to the road, the snow started to melt twelve days earlier than at distances farther away and on May 30 the area nearby the road was snow free. The onset of the snowmelt for the distances $50\,\mathrm{m}$ and $100\,\mathrm{m}$ was similar as for $20\,\mathrm{m}$ but the rate was slower and more area remained snow covered at May 30 as compared to $20\,\mathrm{m}$ from the road edge (Fig. 5b). Areas more than $500\,\mathrm{m}$ from the road edge do not reveal decreasing snow albedo or earlier melt than at

$1000\,\mathrm{m}$ distance (Fig. 5c,d).

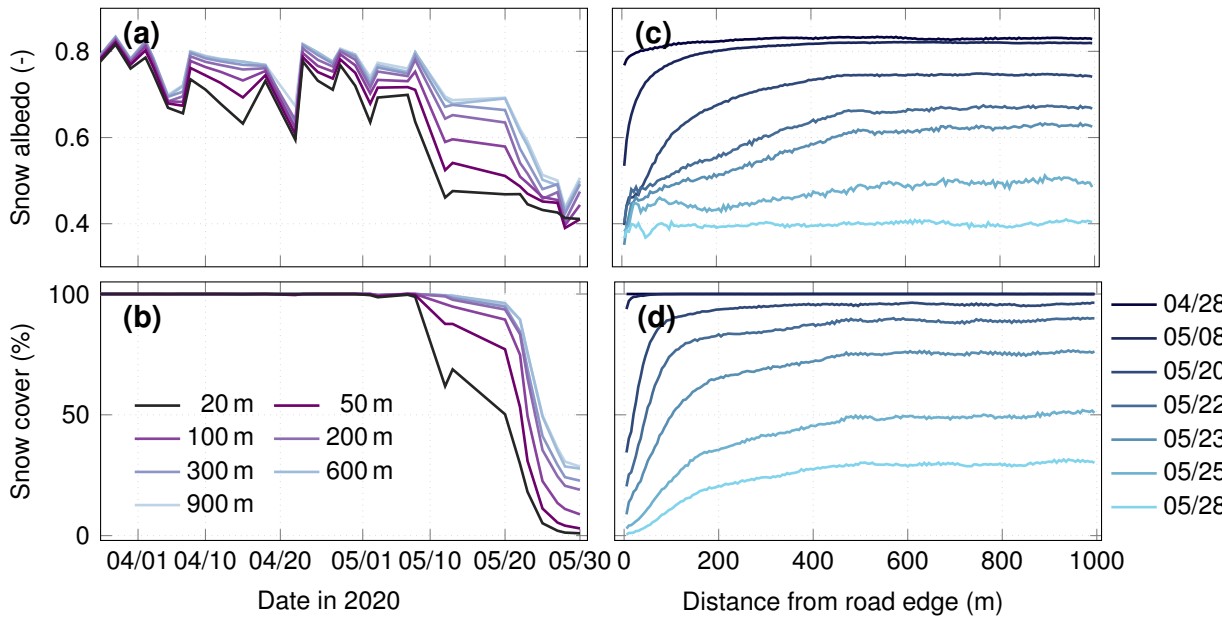

**Figure 5.** Snow albedo and snow cover fraction for Sentinel-2 pixels by distance to the road embankment edge for March 27 to May 30, 2020; left panels: (a) time series of average snow albedo and (b) percentage of snow covered pixels for seven distances; right panels: (c) median snow albedo and (d) percentage of snow covered pixels for seven selected dates by distance to the road edge; please note that the legends of the bottom panels also apply to the top panels.

## 3.3 Spatial extent of early snow free areas

We found that snow close to the road melted earlier. As an example, this is illustrated in Fig. 6a,b, which shows a true color composite and a section of the classified landscape from a Landsat-8 image from May 30, 2018.

The selected images for the late snowmelt period following the start of road construction (2015–2020) clearly showed lower snow cover proximal to the road (Fig. 6c,d). There is also evidence of slightly lower snow cover along the road path prior to construction (2002 and 2006), but the magnitude of this spatial pattern is much stronger following construction. For example, in May 2002 and 2006 the snow cover values were approximately $5\%$ to $10\%$ lower within $100\,\mathrm{m}$ of the future road route than at distances between $100\,\mathrm{m}$ and $1000\,\mathrm{m}$. The snow cover pattern of the year 2015, which corresponds to the period when the quarries were established but not the primary road, exhibited a similar shape as the preceding years.

In 2016, the construction of the road had a strong impact on the snowmelt pattern. Nearly the entire landscape farther from the road was snow covered in the May 6, 2016 image ($80\%$ to $90\%$), while the area in the vicinity of the road was almost snow free. The snowmelt patterns in the imagery from 2017, 2018 and 2020 were similar, both in shape and absolute values of snow cover. The snow cover was lowest at the road edge ($5\%$), increasing abruptly to about $20\%$ ($\pm5\%$) within a distance of $150\,\mathrm{m}$ followed by a more gradual increase to about $30\%$ ($\pm5\%$) snow cover at $500\,\mathrm{m}$. Beyond this distance, the snow cover showed 295 stable values. Although the image from 2019 showed a larger total snow cover than 2017, 2018 and 2020, the road effect was

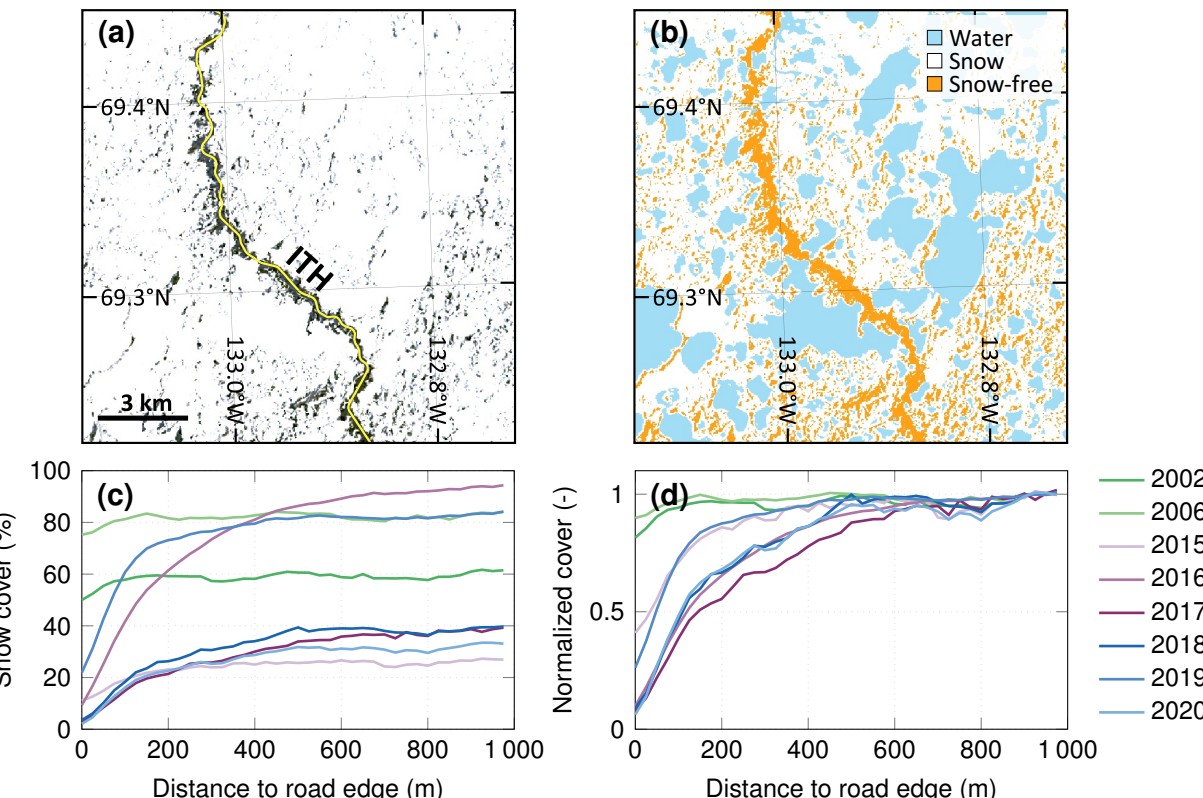

**Figure 6.** (a) True color composite of a partly melted landscape (May 30, 2018, Landsat-8 image), (b) classified landscape with snow covered areas (white), snow free areas (orange) and the water bodies (blue) which were masked using an NDWI threshold on a Landsat-8 image of late summer, (c) presence of snow with distance from ITH derived from Landsat-7 and -8 images using NDSI and (d) relative snow cover as fraction of the value far away from the road (900 m to 1000 m). The images before road construction were from May 17, 2002 and May 19, 2006, during construction from May 13, 2015, May 6, 2016 and May 18, 2017 and after construction from May 30, 2018, May 17, 2019 and May 26, 2020. (a) and (b) depict an example section of the study area (area highlighted in orange in Fig. 1), while (c) and (d) show data from the entire 138 km long road.

similar. The greatest road effect occurred within the first 150 m followed by a gradual increase in snow cover until a distance of 500 m.

To compare the snow cover of the two sides of the road, we subtracted the percentage of snow cover on the road's eastern side from its western side (not shown). Of all the years, only 2016 had a large difference in snow cover fraction between the eastern and western sides of the road. In the 2016 image, the western side showed less snow cover at all distances, with largest differences at distances between 50 m to 500 m. In other years, the difference between the two sides fluctuated around zero (±10 percent points).

The impact of the road on the snow cover becomes more evident when it is compared to undisturbed areas after normalization (Fig. 6d). A normalized snow cover value of 1 indicates that the snow cover at a given distance to the road is comparable to

the undisturbed reference areas. All Landsat images exhibited values close to 1 beyond distances of $600\,\mathrm{m}$ from the road edge, indicating a state of similarity with undisturbed areas. The images from before the construction (2002 and 2006) remained around the value of 1 at closer distances, indicating an undisturbed state. The strongest differences in snow cover compared to the undisturbed region occurred within distances of $150\,\mathrm{m}$ from the road edge for the images from during and after construction. We observed the effects of the road as the onset of a negative trend in normalized snow cover at $400\,\mathrm{m}$ and $600\,\mathrm{m}$ for the years

2015/2019 and 2016–2018/2020, respectively.

## 4 Discussion and outlook

### 4.1 Snow accumulation at the embankment

Snow depth estimates from airborne data along transects across the road revealed enhanced snow accumulation at both sides of the road. This result is in line with other studies based on manual snow surveys (Benson et al., 1975; O'Neill and Burn,

2017). O'Neill and Burn (2017) manually measured the snow depths along transects in tundra and forest along the Dempster Highway on the Peel Plateau (Northwest Territories, Canada) and found an enhanced snow accumulation at the embankment in the tundra landscape. They observed a significant enhancement at a distance of about $5\,\mathrm{m}$ to $15\,\mathrm{m}$ from the side of the road and then a decline with greater distance from the road. Manual snow surveys are very labour intensive. Therefore, the study by O'Neill and Burn (2017) includes only six transects in tundra and a maximum distance of $50\,\mathrm{m}$ to the road. In contrast, the

remote sensing data enabled us to analyse more than 4000 transects of up to $70\,\mathrm{m}$ from the road center. The large number and high density of transects allows the controls on patterns of variation in snow cover to be explored across different spatial scales. In this way, we showed that a gradual increase of snow accumulation reached as far as $36\,\mathrm{m}$ away from the road center. The differences in enhanced snow accumulation compared to O'Neill and Burn (2017) may be attributed to different methodology, precipitation, wind-regimes, vegetation and terrain geometry as well as embankment configuration.

In general, snow accumulation at the embankment can be caused by snow redistribution when the embankment acts as wind break and by snow plowing. While we expect snow plowing to enhance snow accumulation equally on both sides and irrespective of the embankment shape, wind-induced accumulation is likely affected by the local conditions. Snow accumulates preferentially on the lee side of the road (e.g., Benson et al., 1975; Liston and Sturm, 1998). Based on the prevailing southern winds in our study area (Fig. 4c), we expected enhanced snow accumulation towards the north of the road. This was confirmed

by our results (Fig. 4a). The transects in north direction (leeward side) showed more snow accumulation reaching up to $36\,\mathrm{m}$ from the road center as compared to $20\,\mathrm{m}$ at the southern side. At our specific location, the more strongly enhanced snow accumulation at the northern side could also be influenced by the topographic location of the transects. For the road sections studied, the north-facing side was generally located at higher elevations than the south-facing side. At a distance of $60\,\mathrm{m}$ from the road, the elevation towards the road decreased by an average of $1.4\,\mathrm{m}$. Close to the embankment, the slope rapidly changes,

which can cause turbulence and snow deposition.

In the east–west direction, the frequency and average wind speed were similar with slightly stronger winds from the east (Fig. 4c). The similar frequency of wind directions was reflected in the snow accumulation for the transects in east–west direction, which showed similar snow depth patterns on both sides in contrast to north–south facing embankments (Fig. 4a). However, for the transects in western direction, the enhanced snow accumulation reached greater distances from the road center. However, topography may also play an important role as the western side was located at higher elevation compared to the eastern side. Another possibility is that events of very strong winds, which are more frequent when the wind is coming from the east, have a disproportionately large effect on snow redistribution.

For the road sections in our snow accumulation analysis ($4\,km$ of the total $138\,km$ length), we had embankment heights ranging from $0.3\,m$ to $2\,m$. Exploring this variability, we showed that the embankment height influenced the magnitude of the snow accumulation. The embankments that we classified as "high" had, on average, $0.3\,m$ deeper snow as compared to low embankments (Fig. 4b). Higher embankments, locally as high as $12\,m$ (De Guzman et al., 2021), are used on road sections with particularly vulnerable ice-rich permafrost to prevent the underlying permafrost from thawing (Kiggiak - EBA, 2011). However, due to increased snow accumulation, high embankments are one factor within the complex feedback system that could potentially contribute to the degradation of permafrost at the embankment toe. On the other hand, even low embankments increased snow accumulated as compared to distances farther away. Snow plowing may also contribute to enhanced accumulation along the embankment. Typically, low embankments with gentle slopes should facilitate a laminar flow of wind, enabling it to blow the snow away (Lanouette et al., 2015).

It should be noted that our snow accumulation results only represent road sections in the tundra landscape. The snow accumulation along the embankment may be less pronounced further south along the road, where trees reduce winds and inhibit snow drifting (O'Neill and Burn, 2017). Notwithstanding these limitations, these results give us a better understanding of the spatial distribution of snow next to a permafrost road embankment in a tundra environment.

Snow accumulation was identified as one of the key factors contributing to permafrost degradation and increased thaw depths along infrastructure (Park et al., 2015; Schneider von Deimling et al., 2021). The insulating properties of deep snow lead to warmer ground temperatures in winter and thus increases the thickness of the active layer in the following summer (Gouttevin et al., 2012). Moreover, increased snow accumulation contributes to higher soil moisture and pond formation and, thus, a delayed soil refreezing in autumn (Zhang, 2005).

Due to the limited data availability, we studied snow accumulation patterns at a single date (April 10, 2019). According to the ERA5 dataset, April 2019 had $0.1\,m$ less than average snow depth but rapid snowmelt had not started by the time of data acquisition. Furthermore, the spatial pattern of snow distribution remains stable from year to year (Sturm and Wagner, 2010). Therefore, the map of snow distribution and snow depth we produced could be integrated into permafrost degradation and blowing snow models and contribute to their validation and improvement.

## 4.2 Snow albedo decrease

We provided the first Sentinel-2 derived snow albedo time series along the ITH for the year 2020. Remotely-sensed snow albedo values were similar across all distances from the road throughout winter, until early April (Fig. 5). This indicates that

frequent new snowfalls or snow redistributed by winds mask the deposited dust. Moreover, the frozen state of the road surface and the protective layer of snow or ice on the road may shield the dust from mobilization, potentially leading to increased production in spring relative to fall and winter. However, it is important to note that we currently lack ground-based data (snow samples and albedo measurements) to confirm this hypothesis empirically. After April 5, the remote sensing data indicates that the snow albedo was lower at distances close to the road and higher at distances farther away. This may be due to dust accumulation on the snow surface as snow melts and settles. We detected road related effects on the snow albedo up to $500\,\mathrm{m}$ from the road. In contrast, snow albedo at $600\,\mathrm{m}$ and $900\,\mathrm{m}$ from the road showed similar values and temporal patterns across whole study period and therefore indicated little or no effect of road-dust at these distances. Moreover, the snow cover fraction decreased earlier at closer distances to the road as compared to farther away. At $20\,\mathrm{m}$ distance, the melt-off started May 8, 2020 while it started twelve days later at $600\,\mathrm{m}$ distance. This finding is in line with Benson et al. (1975) who described a two-week premature snowmelt along roads in Prudhoe Bay region, Alaska, due to dust lowering the snow albedo.

### 4.3  Spatial extent of early snow free areas

We examined the snowmelt patterns along the road using the NDSI derived from Landsat data. Even though snow accumulates near the road, these areas are experiencing spring-melt and snow free conditions before areas distal to the road (Fig. 6). Most of the enhanced snow accumulation is within $30\,\mathrm{m}$ as shown in Fig. 4, which is also the spatial resolution of the Landsat bands used for calculating the NDSI. Therefore, even if the snow in the absolute vicinity of the road persists longer because of greater snow depth, it may not be visible in the images because of the spatial resolution of the utilized sensor. However, knowledge of the field conditions indicates that snow in these areas melts early.

The results are in line with other similar studies showing earlier snowmelt next to a gravel road on permafrost in Prudhoe Bay, Alaska (Bergstedt et al., 2022). However, unlike their study we could not identify any considerable effect when classifying the pixels depending on the side of the road. A possible explanation is, that the majority of the road is located in north–south direction and the prevailing wind direction in winter 2018–2019 was from the south (Fig. 4f). The eastern and western winds are similar in wind speed and occurrence which may explain why we did not find substantial differences between the sides. Furthermore, we classified the landscape west and east of the road and did not consider short sections of the road which are in east–west direction.

Other studies have identified road dust as the main cause of the early melt-off in spring (Everett, 1980; Walker and Everett, 1987). Dust loading on snow leads to decreased snow albedo and, thus, increased absorption of solar energy, inducing snowmelt. Everett (1980) showed that most dust falls within a distance of $300\,\mathrm{m}$ with a logarithmic decrease with distance. This finding is in accordance with the spatial pattern of snowmelt found in our work (Fig. 5b,d and 6c,d). This was especially clear in the images from the years 2016 and 2019, representing a high snow cover farther away from the road (90% and 80%), while the area close to the road was already mostly snow free.

We detected earlier snowmelt at distances up to $600\,\mathrm{m}$ from the road depending on the observation date, which is a greater distance than in the study by Benson et al. (1975). The main snowmelt extent in our study was within a zone of $200\,\mathrm{m}$ whereas Benson et al. (1975) noted that the early snowmelt occurred primarily within a $100\,\mathrm{m}$ zone (along roads in the Prudhoe Bay

region). One possible explanation for the differences in snowmelt may be the misclassification of the NDSI when the snow is contaminated with dust and thus falsely classified as snow free. However, the reliability of the NDSI given different spectral characteristics of the snow was examined in Kulkarni et al. (2002). They found that the NDSI values for all types of snow, such as fresh, clean, patchy, wet and contaminated were substantially different from snow free areas. A second explanation for the differences in snowmelt extent may be different wind speeds, dust particle size and snow clearance practices. Furthermore, our method may be more sensitive to small effects as we analyse thousands of satellite pixels. On the other hand, Bergstedt et al. (2022) analysed Sentinel-2 data at Prudhoe Bay, Alaska and found that some areas up to $5\,km$ from roads or pipelines showed earlier snowmelt. However, they also found the major effects within the first $100\,m$.

We found reduced snow cover along the road also in the images before road construction. This can be attributed to the road being constructed as much as possible over elevated well-drained terrain, which typically exhibits lower snow accumulation than lower lying areas and topographic depressions. Moreover, the vegetation in the region affects the timing of snowmelt, with dwarf birch-dominated areas experiencing earlier snowmelt compared to other vegetation types (Wilcox et al., 2019). Dense dwarf birch canopies are often associated with elevated terrain or slopes rather than depressions (Grünberg et al., 2020). These factors can collectively contribute to a region experiencing a more frequent and earlier absence of snow compared to other areas within the landscape.

Permafrost is affected by the early snowmelt in multiple ways. Auerbach et al. (1997) found the deepest thaw next to the Dalton Highway (Alaska) and a consistent decrease with distance from the road. They attributed the increased thaw depth close to the road to the earlier exposure to solar radiation. The melt-off can start up to 14 days before the general melt-off (Walker and Everett, 1987), which can affect the underlying vegetation. Vegetation phenology and start of the season are significantly correlated with the last day of snow cover (Zeng and Jia, 2013). Some plants are more favored by early snowmelt than others, as noted in Auerbach et al. (1997). They highlighted the close relationship between early snowmelt and vascular plant growth, deciduous shrub and sedge flowering. Tall shrubs may act as a windbreak, increasing dust deposition and snow accumulation, leading to more soil nutrients, higher soil temperatures and increased active layer thickness (Gill et al., 2014). Our results indicate, that dust deposition is the main driver of the early snowmelt and the lower snow albedo at distances closer to the road. The affected footprint may be larger than earlier considered and even if there is currently no evidence for permafrost degradation at such distances, these seemingly undisturbed areas may be affected in the future.

## 4.4 Conclusion

In this study, we utilized remote sensing techniques to show enhanced snow accumulation, reduced snow albedo and the extent of earlier snowmelt following the construction of the Inuvik to Tuktoyaktuk Highway. With a new, high resolution snow depth raster, our study confirms that the presence of the road significantly affects snow accumulation, resulting in up to $0.8\,m$ more snow compared to adjacent undisturbed areas and enhanced snow depth up to $36\,m$ from the road. High embankments result in deeper snow, but also low embankments with gentle slopes contribute to enhanced snow accumulation, which potentially contributes to permafrost degradation. In addition to increased snow depth at the embankment, we observed a decrease in snow albedo and earlier snowmelt near the road, which may have significant implications for permafrost stability and ecosystem

dynamics. The impact of the highway on snow albedo reached up to $500\,\mathrm{m}$ and the extent of earlier snowmelt up to $600\,\mathrm{m}$ from the road, highlighting the far-reaching influence of gravel roads on the surrounding tundra environments. Understanding the effects of snow–road interactions in permafrost areas is crucial for current management practices and future development of mitigation strategies. Our results contribute to a better understanding of road effects on snow, a potential driver of permafrost degradation. These snow effects are most pronounced locally, but we show effects up to seemingly undisturbed areas at greater distance to the road. Furthermore, we recommend increased monitoring of dust deposition along Arctic highways and further research to calibrate remote sensing techniques which show significant potential for gaining a better understanding of the impacts of road infrastructure on tundra vegetation and the surrounding environment. These efforts will contribute to informed mitigation strategies and decision making regarding infrastructure planning, development and maintenance during a time of unprecedented climate warming.

*Data availability.* The snow covered DEM and snow depth product are available on PANGAEA (Hammar et al., 2023).

*Author contributions.* J. H. designed the study and led the manuscript preparation. I. G. and J. B. initiated and supervised the study and advised on the methodology. I. G. created the figures with contributions by J. H.. All authors contributed to refining the study design and analyses for publication, interpreting the results and collectively contributed to the manuscript's text.

*Competing interests.* The authors declare that they have no competing interests.

*Acknowledgements.* We thank the AWI IceBird team for the 2019 overflight over the ITH, providing us with essential ALS data and Timothy Ensom for pointing us to related research by the Northern Territory Geological Survey (NTGS) and providing valuable input. Part of this work (Jennika Hammar/Inge Grünberg) was funded by Helmholtz Imaging, a platform of the Helmholtz Incubator on Information and Data Science (grant number: ZT-I-PF-4-001). The research was conducted under the NWT science licence No. 17024. The authors acknowledge that this study was conducted in the Inuvialuit settlement region in the western Canadian Arctic. We also thank the TVC research station team for scientific discussions, assistance with field work and long-term data collection and the Aurora Research Institute (ARI) for logistical support.

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
