# Peer review of "Snow accumulation, albedo and melt patterns following road construction on permafrost, Inuvik-Tuktovaktuk Highway, Canada"

_The Cryosphere, 2023_

## Author Comment (AC1)

**Replies to referee comments on:**

**Snow accumulation, albedo and melt patterns following road construction on permafrost, Inuvik-Tuktoyaktuk Highway, Canada**

J. Hammar, I. Grünberg, S. V. Kokelj, J. van der Sluijs and J. Boike
*The Cryosphere,*

*RC: Referee's Comment*,     AR: Authors' Response,     ☐ Manuscript Text

**Referee #1**

*This study looks to quantify snow conditions along the newly constructed Inuvik-Tuktoyaktuk Highway, a gravel highway extending 138 km along the tundra-taiga ecotone in the Mackenzie Delta uplands. The highway was constructed entirely on ice-rich permafrost, and is therefore sensitive to the underlying permafrost conditions, which have been observed rapidly warming in this region. The study analyzes the localized influence of the highway embankment on snow accumulation and the effects of road dust on late season albedo, snowmelt timing and extent. Through a combination of several platforms (Airborne Lidar, Landsat 7+8, and Sentinel-2) the authors adequately apply widely accepted remote sensing methods to undertake a novel analysis of the effect of all-season tundra roads on snow conditions. The manuscript is well written, clear and well structured, and acceptable for publication in TC.*

**Response to Referee #1**

We are very grateful for the constructive comments and suggestions provided by the referee that have significantly improved our manuscript. Please find our responses and relevant changes to comments below (referee comment in gray italic and authors response in normal font).

**Minor comments:**

*RC:*    *Line 29. The drifting of snow is widely studied in this study region, specifically at TVC. Perhaps the authors should include reference to snow accumulation beyond infrastructure.*

AR:    In line 64, we refer to the studies by Wilcox et al. (2019) and Walker et al. (2021), which use drone imagery to examine snow depth and timing of snowmelt. Furthermore, we added results of previous studies related to the influence of shrubs on snow albedo and snowmelt timing to the introduction in the revised version:

> As shrubs act as a windbreak, increased shrub growth alters the snow thickness [...] In addition to this potential warming effect, shrub branches also protrude the snow surface in spring, which lowers the albedo and may lead to earlier snowmelt and thus a longer summer season (Marsh et al., 2010; Wilcox et al., 2019).

*RC:*    *Line 52. What on the influence of shading of tall shrubs on reducing incoming radiation and decreasing the amount of energy used to warm the soil? And what are the effects of soil moisture characteristics on the thermal regimes of the underlying ground cover? There are numerous studies mentioning this effect. May be worth mentioning to provide the readers with a full understanding of the shrub-snow-soil feedback system.*

AR: We agree that it is worth including the shading effect of shrubs during summer and revised the paragraph as follows:

>  In addition to this potential warming effect, shrub branches also protrude the snow surface in spring, which lowers the albedo and may lead to earlier snowmelt and thus a longer summer season (Marsh et al., 2010; Wilcox et al., 2019). In contrast, during summer, the presence of shrubs reduces the incoming radiation, leading to cooler soils and potentially shallower active layer thickness (Lawrence and Swenson, 2011; Myers-Smith and Hik, 2013). However, the reduced availability of light, water and nutrients has a negative impact on mosses growing below the shrubs (Gill et al., 2014). This, together with heavy dust deposition, results in a decline in moss and lichen cover, which may decrease thermal insulation of the ground in summer, further amplifying soil warming (Raynolds et al., 2014).

We further included the effects of soil moisture characteristics on the thermal regimes of the underlying ground cover in the following paragraph:

> Process-based land surface models showed [...] are most strongly affected by early season snow accumulation (Park et al., 2015). Furthermore, increased snow accumulation serves as an additional source of water when it melts, contributing to higher soil moisture and pond formation along the embankment in the spring. The elevated soil moisture, in turn, impacts the subsurface thermal regime by delaying soil refreezing due to the release of latent heat (Hinkel et al., 2001; Zhang, 2005).

RC: *Line 64. Wilcox et al. also included mapping ultra-high resolution snowmelt timing using optical drone imagery (and along the proximity of the primary study site for this manuscript).*

AR: We have revised the sentence to also include the snowmelt timing from Wilcox et al.

> Drone imaging have been successfully used to determine snow depth distribution and snowmelt timing in the Trail Valley Creek (TVC) watershed, NWT using structure from motion techniques (Wilcox et al., 2019; Walker et al., 2021).

RC: *Line 94. Add "The" before ITH.*

AR: Done.

> The ITH is located within [...]

RC: *Line 127. I am curious about the "1.41m" search radius. What is the reasoning behind this specific value?*

AR: We used the default search radius formula in the Points2Grid approach which is defined as: $search\ radius = grid\ resolution \cdot \sqrt{2}$, we have revised the sentence to include the formula:

> We interpolated the ground-classified dataset with inverse distance weighting (IDW) using the  Points2Grid approach integrated into PDAL , with the default circular neighborhood search radius of $grid\ resolution \cdot \sqrt{2}$, resulting in a radius of 1.41m.

RC: *Line 134. The absence of in-situ snow depths along this 4km study site of TVC is unfortunate, however the authors have developed a clever method for testing the accuracy of their DTM snow depths.*

AR: We were fortunate to have two datasets with such a small time gap that we could use the road as a reference for validation. If the datasets had been spaced further apart in time, we would not have been able to rely on the road as a reference due to ground subsidence and settlement of the road material after construction.

RC: *Line 238. What was the average snowfall for these years at this study site? It would be useful to compare the effects of the highway embankment accumulation zones to the natural snowfall, or snow accumulation of the region*

AR: We appreciate the reviewer's valuable suggestion and incorporated a comparison between the snow accumulation within the highway embankment zones and the region's average snow depth in the revised manuscript using the nine nearest pixels surrounding the transects of the ERA5-Land reanalysis long-term dataset (Muñoz Sabater, J., 2019). In general, April 2019 $(0.5\,\text{m})$ has less snow than the multi-year average of that date $(0.6\,\text{m})$ but the snow depth is well within the spread of other years and the rapid snow depth decline only started one month after our overflight (Fig. R1). The ERA5 snow depth matches well with the average which we observed at the north and west of the road, while we observed lower snow depth at the south and east sides. This is likely due to the specific topography at the studied road section. We added the additional method, a short info in the results, and some sentences in the discussion section:

Method:

> We used ERA5 snow depth data from 1950–2022 (Muñoz Sabater, J., 2019) with a horizontal resolution of 0.1°x0.1° to assess whether or not April 2019 was representative for typical spring conditions. To achieve this, we calculated the daily average snow depth for each year from the nine nearest pixels surrounding the transects.

Results:

> The average snow depth at the northern and western sides of the embankment matches well with the regional average snow depth obtained from the ERA5 snow depth product of 0.5 m at the day of our data acquisition (2019-04-10).According to the ERA5 dataset, the multi-year average snow depth at this date is 0.6 m (1950–2022). Thus the snow depth of April 2019 was below average, but 25% of the years had even less snow and rapid snowmelt did not start until one month later.

Discussion:

> As compared to other years, April 2019 had less than average snow depth. However, rapid snowmelt had not started by the time of data acquisition.

[Figure]

Figure R1: Snow depths of 1950–2022 derived from ERA5 reanalysis data with the days of the year 200-365 from 2018 in blue and the year 2019 highlighted in red.

RC: *Figure 4. I like this figure as it provides a clean and easy to follow overview of the snow accumulation patterns based on slope and aspect. However I have two comments. First, the authors need to include a picture of the highway and snow embankments. This reviewer is familiar with the study location, but others may not be. Second, where is this data sourced from? It if ECCC or the nearby TVC research station? The authors need to cite the source.*

AR: We added the reference to the wind data and changed the caption to Figure 4 and added two pictures of the highway:

> (c) shows the predominant wind direction  for the Trail Valley October 2018 – April 2019 (climate identifier: 220N005, current station operator: Environment and Climate Change Canada - Meteorological Service of Canada) (Environment and Climate Change Canada, 2023).

[Figure]

Figure R2: Inuvik-Tuktoyaktuk Highway (ITH) on August 24, 2022 (a) and after the first snowfall on September 23, 2021 (b).

RC: *Section 3.2. Snow Albedo decrease: This makes sense as it corresponds to the timing of the (enhanced) snowmelt with proximity to the road. I was hoping the study would have talked about increasing albedo over the snow accumulation period. But I have also noticed in similar studies that the dust produced during the winter (during non-construction years) is minimal until the*

*snow begins to melt on the road surface, and dust is free to be redistributed. The authors do a good job in the discussion describing why this may be so (lack of dust availability due to the snow-capping of the road surface).*

AR: Your comment about increased snow albedo over the snow accumulation period is a great point which we may consider in future research. Your insights into dust production during the winter and its connection to snowmelt are noted, and we are glad you found our discussion informative. Moreover we revised the following sentence to also include the potential increased dust production in spring:

> Moreover, a protective layer of snow or ice on the road may shield the dust from mobilization, potentially leading to increased production in spring relative to fall and winter. However, it is important to note that we currently lack ground-based data (snow samples and albedo measurements) to confirm this hypothesis empirically.

*RC: Figure 5, a. What are the "seven distances", it was not clear in the panel*

AR: Figure 5a is sharing the x-axis and legend with 5b, we understand that this is not intuitive and added this information in the caption in the revised version of the paper.

*RC: Discussion. The authors did a good job here addressing several key issues discussed in their results section relating to snow clearance practices and snow accumulation, dust availability and albedo, and the effects of the highway orientation on all three snow characteristics discussed with. One question, did the authors look at spatial patterns of snowmelt timing with relation to vegetation cover? Perhaps an area of future study using higher spatial and temporal resolution (such as the drones mentioned early in the methodology section).*

AR: Your question about the spatial patterns of snowmelt timing in relation to vegetation cover is interesting and aligns with the broader context of our research. It is also already subject of related studies at the TVC site, such as Wilcox et al. (2019). While this aspect was beyond the scope of our current study, we appreciate your suggestion and agree that it is a valuable area for future study. We are currently looking at patterns of snow accumulation in more detail and hope to follow up on this topic soon.

*RC: Line 390. Wilcox et al. (2019) looked at snowmelt patterns across the entire 2016 snowmelt period in the TVC study area with relation to vegetation and permafrost development. This may be a more applicable and recent study to include in the discussion. Could be worth citing here.*

AR: We agree with the comment and have revised the paragraph to further include the influence of vegetation type on the snowmelt timing:

> This can be attributed to the road being constructed in interfluvial regions and at elevated terrain, which typically exhibits lower snow accumulation than lower lying areas and topographic depressions. Moreover, the vegetation in the region affects the timing of snowmelt, with dwarf birch-dominated areas experiencing earlier snowmelt compared to other vegetation types (Wilcox et al., 2019). Dense dwarf birch canopies are often associated with elevated terrain rather than depressions. These factors can collectively contribute to a region experiencing a more frequent and earlier absence of snow compared to other areas within the landscape.

*RC: Acknowledgements. The authors should include reference to their NWT science license number. What about contributing acknowledgement to the Trail Valley Creek Research Station, the primary study site?*

AR: We appreciate the suggestion and have revised our acknowledgement:

Part of this work [...]. The research was conducted under the NWT science licence No. 17024. The authors acknowledge that this study was conducted in the Inuvialuit settlement region in the western Canadian Arctic. We also thank the TVC research station team for scientific discussions, assistance with field work and long-term data collection and the Aurora Research Institute (ARI) for logistical support.

**A: Additional comments from the authors**

In the editor's initial decision, we were recommended to double check the location of the treeline in Figure 1, which appeared to be too far north in relation to the TVC research station. The definition and precise location of the treeline varies strongly across different scientific publications and studies. For the Mackenzie Delta region in particular, a large number of studies displays different tree lines (Eaton et al., 2001; Raynolds et al., 2019; Brandt, 2009; Burn and Kokelj, 2009; Palmer et al., 2012; Fraser et al., 2014; O'Neill et al., 2023). An important study by Antonova et al. (2019) estimates tree height and forest properties using TanDEM-X data in small forest patches located at the northern edge of the treeline zone in the Canadian Arctic, specifically including TVC. In this study, the term "treeline zone" is utilized to refer to the transition area between boreal forest and tundra, implying that the change is not abrupt, and a gradient of tree densities and heights can be observed within this zone. Therefore, in the revised version of our manuscript we refer to the region as "treeline zone".

**References**

Antonova, S., Thiel, C., Höfle, B., Anders, K., Helm, V., Zwieback, S., Marx, S., and Boike, J.: Estimating tree height from TanDEM-X data at the northwestern Canadian treeline, Remote Sensing of Environment, 231, https://doi.org/10.1016/j.rse.2019.111251, 2019.

Brandt, J.: The extent of the North American boreal zone, Environmental Reviews, 17, 101–161, https://doi.org/10.1139/A09-004, 2009.

Burn, C. R. and Kokelj, S. V.: The environment and permafrost of the Mackenzie Delta area, Permafrost and Periglacial Processes, 20, 83–105, https://doi.org/10.1002/ppp.655, 2009.

Eaton, A. K., Rouse, W. R., Lafleur, P. M., Marsh, P., and Blanken, P. D.: Surface Energy Balance of the Western and Central Canadian Subarctic: Variations in the Energy Balance among Five Major Terrain Types, Journal of Climate, 14, 3692–3703, https://doi.org/10.1175/1520-0442(2001)014<3692:SEBOTW>2.0.CO;2, 2001.

Environment and Climate Change Canada: Historical data Trail Valley, Northwest Territories, URL: https://climate.weather.gc.ca/, accessed: 2023-08-12, 2021.

Fraser, R. H., Lantz, T. C., Olthof, I., Kokelj, S. V., and Sims, R. A.: Warming-Induced Shrub Expansion and Lichen Decline in the Western Canadian Arctic, Ecosystems, 17, 1151–1168, https://doi.org/10.1007/s10021-014-9783-3, 2014.

Gill, H. K., Lantz, T. C., O'Neill, B., and Kokelj, S. V.: Cumulative impacts and feedbacks of a gravel road on shrub tundra ecosystems in the Peel Plateau, Northwest Territories, Canada, Arctic, Antarctic, and Alpine Research, 46, 947–961, https://doi.org/10.1657/1938-4246-46.4.947, 2014.

Hinkel, K., Paetzold, F., Nelson, F., and Bockheim, J.: Patterns of soil temperature and moisture in the active layer and upper permafrost at Barrow, Alaska: 1993–1999, Global and Planetary Change, 29, 293–309, https://doi.org/10.1016/S0921-8181(01)00096-0, 2001.

Lawrence, D. M. and Swenson, S. C.: Permafrost response to increasing Arctic shrub abundance depends on the relative influence of shrubs on local soil cooling versus large-scale climate warming, Environmental Research Letters, 6, 045 504, https://doi.org/10.1088/1748-9326/6/4/045504, 2011.

Marsh, P., Bartlett, P., MacKay, M., Pohl, S., and Lantz, T.: Snowmelt energetics at a shrub tundra site in the western Canadian Arctic, Hydrological Processes, 24, 3603–3620, https://doi.org/10.1002/hyp.7786, 2010.

Muñoz Sabater, J.: ERA5-Land hourly data from 1950 to present. Copernicus Climate Change Service (C3S) Climate Data Store (CDS), https://doi.org/10.24381/cds.e2161bac, accessed: 2023-09-19, 2019.

Myers-Smith, I. H. and Hik, D. S.: Shrub canopies influence soil temperatures but not nutrient dynamics: An experimental test of tundra snow–shrub interactions, Ecology and Evolution, 3, 3683–3700, https://doi.org/10.1002/ece3.710, 2013.

O'Neill, H. B., Smith, S. L., Burn, C. R., Duchesne, C., and Zhang, Y.: Widespread Permafrost Degradation and Thaw Subsidence in Northwest Canada, Journal of Geophysical Research: Earth Surface, 128, e2023JF007 262, https://doi.org/10.1029/2023JF007262, e2023JF007262 2023JF007262, 2023.

Palmer, M., Burn, C., Kokelj, S., and Allard, M.: Factors influencing permafrost temperatures across tree line in the uplands east of the Mackenzie Delta, 2004-2010, Canadian Journal of Earth Sciences, 49, 877–894, https://doi.org/10.1139/e2012-002, 2012.

Park, H., Fedorov, A. N., Zheleznyak, M. N., Konstantinov, P. Y., and Walsh, J. E.: Effect of snow cover on pan-Arctic permafrost thermal regimes, Climate Dynamics, 44, 2873–2895, https://doi.org/10.1007/s00382-014-2356-5, 2015.

Raynolds, M. K., Walker, D. A., Ambrosius, K. J., Brown, J., Everett, K. R., Kanevskiy, M., Kofinas, G. P., Romanovsky, V. E., Shur, Y., and Webber, P. J.: Cumulative geoecological effects of 62 years of infrastructure and climate change in ice-rich permafrost landscapes, Prudhoe Bay Oilfield, Alaska, Global Change Biology, 20, 1211–1224, https://doi.org/10.1111/gcb.12500, 2014.

Raynolds, M. K., Walker, D. A., Balser, A., Bay, C., Campbell, M., Cherosov, M. M., Daniëls, F. J., Eidesen, P. B., Ermokhina, K. A., Frost, G. V., Jedrzejek, B., Jorgenson, M. T., Kennedy, B. E., Kholod, S. S., Lavrinenko, I. A., Lavrinenko, O. V., Magnússon, B., Matveyeva, N. V., Metúsalemsson, S., Nilsen, L., Olthof, I., Pospelov, I. N., Pospelova, E. B., Pouliot, D., Razzhivin, V., Schaepman-Strub, G., Šibík, J., Telyatnikov, M. Y., and Troeva, E.: A raster version of the Circumpolar Arctic Vegetation Map (CAVM), Remote Sensing of Environment, 232, 111 297, https://doi.org/10.1016/j.rse.2019.111297, 2019.

Walker, B., Wilcox, E. J., and Marsh, P.: Accuracy assessment of late winter snow depth mapping for tundra environments using structure-from-motion photogrammetry, Arctic Science, 7, 588–604, https://doi.org/10.1139/as-2020-0006, 2021.

Wilcox, E. J., Keim, D., de Jong, T., Walker, B., Sonnentag, O., Sniderhan, A. E., Mann, P., and Marsh, P.: Tundra shrub expansion may amplify permafrost thaw by advancing snowmelt timing, Arctic Science, 5, 202–217, https://doi.org/10.1139/as-2018-0028, 2019.

Zhang, T.: Influence of the seasonal snow cover on the ground thermal regime: An overview, Reviews of Geophysics, 43, https://doi.org/10.1029/2004RG000157, 2005.

---

## Author Comment (AC2)

**Replies to referee comments on:**

**Snow accumulation, albedo and melt patterns following road construction on permafrost, Inuvik-Tuktoyaktuk Highway, Canada**

J. Hammar, I. Grünberg, S. V. Kokelj, J. van der Sluijs and J. Boike
*The Cryosphere,*
* * *
*RC: Referee's Comment*,     AR: Authors' Response,     ☐ Manuscript Text

**Referee #2**

*The manuscript is really well written, the figures are of high quality and the topic is interesting and well presented. The methodology is sound and the selected data sources are appropriate for the analysis. This manuscript has the potential to be a good contribution to the literature.*

*My main comment to the authors would be that the novelty of the work could be improved/highlighted more. Many of the points made in the discussion are well established in previous studies (as pointed out by the authors). However, I do believe that that this manuscript has many unique parts that just could be highlighted a bit more. The following points are suggestions of how this paper could be improved and the full potential of this study could be unlocked. It may not be necessary to do all of these, but including some may improve the uniqueness of this study.*

**Response to referee #2**

We thank you for taking the time to review our manuscript and for providing helpful and constructive feedback on our work. We are confident that our revised manuscript has significantly improved due to your suggestions and look forward to its possible publication in the TC. Please find our responses and relevant changes based on your comments below (referee comment in gray italic and authors response in normal font).

RC:   *A unique feature of this study is the high resolution DEM and the derived high resolution snow depth data. I believe it would be a great addition if this would be discussed more, especially analysed in from a spatial point of view. The authors limit themselves with sticking to selected transects.*

AR:   We totally agree with the reviewer that we should give more attention to this high resolution DEM, we have revised our manuscript and included it in the abstract and in the conclusion. Regarding the concern about selected transects, we want to clarify that our chosen transects, spaced at 1 m intervals, actually cover the entire highway section for which we have data. We acknowledge the need for improved clarity in our manuscript and have addressed this in the revision:

Abstract:

>  With a new, high resolution snow depth raster, we quantified the snow accumulation at road segments in the Trail Valley Creek area using digital elevation model differencing.

Methods:

> We created 140 m long transects (n = 4026) perpendicular to the road every $1\,\text{m}$ over the ITH centerline using GRASS GIS . This ensured comprehensive coverage of the entire section of highway from which we have data. Subsequently, we extracted the snow depth values.

Conclusion:

> With a new, high resolution snow depth raster, Oour study confirms that the presence of the road significantly affects snow accumulation, resulting in [...]

RC: *In the introduction the authors highlight the snow accumulation leading to water ponding and the importance of this to vegetation (and permafrost conditions). This is neglected in the analysis of the data and the discussion. The authors also introduce the NDWI within the method section. It would be relatively easy to expand the study to include an analysis of water ponding after snow melt along the highway.*

AR: In the introduction, we highlighted the significance of snow accumulation leading to water ponding and its relevance for permafrost degradation. We looked into this topic in detail before we decided not to include this part in our presented study. As part of the first author's master's thesis (Hammar, 2022), the NDWI was initially used to analyze the ponds along the highway. However, she observed a common occurrence of negative NDWI values in the small ponds, contrary to the expected positive values for water bodies. This behavior of water pixels in the visible spectrum is typical in high-latitude regions, characterized by a low sun zenith angle and shallow water bodies that facilitate phenomena such as sun glint, turbidity, and lake bottom reflectance (Muster et al., 2013). Consequently, we concluded that the NDWI may not be suitable for delineating water bodies along the highway, especially given that the highway is relatively new, and the ponds may not yet be large enough to be captured at the spatial resolution of the satellite imagery. Nevertheless, we acknowledge the importance of studying water ponding after snowmelt along the highway. We believe that the use of the NIR band alone could be a possible way to delineate ponds. However, developing a reliable and accurate method for this purpose is beyond the scope of our current study. Another option may be higher resolution airborne datasets. This topic remains of interest to us and we intend to explore it in future research.

RC: *It would be interesting to include a spatial analysis of the albedo. Maybe for selected dates. If no analysis, maybe at least a map.*

AR: Thank you for your suggestion. Since our primary focus in this study was the road, we have presented a spatial analysis over several distances from the road in Figure 5. For every distance we calculated the average snow albedo in order to reduce other effects such as vegetation. We believe that this method provides more valuable information than an albedo map for our specific research objectives.

RC: *The relatively recent creation of the highway allows remote sensing analysis prior to the existence of the infrastructure. The analysis of snow patterns prior and post the existence of the highway is very interesting, what about things like ponding and NDVI in the area of the highway. This is also (partially) mentioned in the introduction but not further explored in the paper.*

AR: Although we initially studied NDWI and NDVI changes after road construction, we decided not to include this work in the current study. We did not succeed to quantify ponding due to the small pond sizes and the pond reflection characteristics. Regarding the analysis of NDVI we have two major reasons not to include it in our current study:

First, the limitation stems from the availability of Sentinel-2 data going back only to 2015, which coincides with the construction period of the road. Therefore, the high resolution of Sentinel-2 images is not available for the pre-construction period. Landsat, on the other hand, has a lower spatial resolution of only $30\,\text{m}$, which is too coarse to capture areas close to the road as mixed pixels including parts of the gravel road need to be excluded. However, we recognize the potential utility of Landsat data in the long term, especially as the road ages and the vegetation damage caused by construction recovers.

Second, NDVI values as a measure of vegetation biomass measured next to a gravel road may be misleading. Road dust leads

to reduced summer NDVI values due to the dust blocking the spectral characteristics of the underlying vegetation (Ackerman and Finlay, 2019). Given the findings and cautions raised in Ackerman and Finlay (2019), we decided not to include our NDVI analysis in the current study.

Still, we agree that vegetation change and water ponding following road construction are very interesting topics that could be studied in the future, potentially utilizing other high-resolution datasets for a more comprehensive examination.

**Minor points:**

*RC:* *Where was the wind data measured. Please include the source in the source in the caption.*

AR: We included the data source in the caption of the revised manuscript:

> (c) shows the predominant wind direction  for the Trail Valley October 2018 – April 2019 (climate identifier: 220N005, current station operator: Environment and Climate Change Canada - Meteorological Service of Canada) (Environment and Climate Change Canada, 2023).

*RC:* *Line 374. "We detected earlier snowmelt at distances up to 600 m from the road depending on the observation date, which is greater than in previous studies". Bergstedt et al. (2022) (cited by the authors) report a distance of up to 5 km.*

AR: You are correct, Bergstedt et al. (2022) report an effect of a road on snowmelt up to $5\,\mathrm{km}$ distance. We corrected our statement in the revised version of our paper:

> We detected earlier snowmelt at distances up to 600 m from the road depending on the observation date, which is  a greater distance than in the study by Benson et al., 1975.

**A: Additional comments from the authors**

In the editor's initial decision, we were recommended to double check the location of the treeline in Figure 1, which appeared to be too far north in relation to the TVC research station. The definition and precise location of the treeline varies strongly across different scientific publications and studies. For the Mackenzie Delta region in particular, a large number of studies displays different tree lines (Eaton et al., 2001; Raynolds et al., 2019; Brandt, 2009; Burn and Kokelj, 2009; Palmer et al., 2012; Fraser et al., 2014; O'Neill et al., 2023). An important study by Antonova et al. (2019) estimates tree height and forest properties using TanDEM-X data in small forest patches located at the northern edge of the treeline zone in the Canadian Arctic, specifically including TVC. In this study, the term "treeline zone" is utilized to refer to the transition area between boreal forest and tundra, implying that the change is not abrupt, and a gradient of tree densities and heights can be observed within this zone. Therefore, in the revised version of our manuscript we refer to the region as "treeline zone".

**References**

Ackerman, D. E. and Finlay, J. C.: Road dust biases NDVI and alters edaphic properties in Alaskan arctic tundra, Scientific Reports, 9, 1–8, https://doi.org/10.1038/s41598-018-36804-3, 2019.

Antonova, S., Thiel, C., Höfle, B., Anders, K., Helm, V., Zwieback, S., Marx, S., and Boike, J.: Estimating tree height from TanDEM-X data at the northwestern Canadian treeline, Remote Sensing of Environment, 231, https://doi.org/10.1016/j.rse.2019.111251, 2019.

Bergstedt, H., Jones, B. M., Walker, D. A., Peirce, J. L., Bartsch, A., Pointner, G., Kanevskiy, M. Z., Raynolds, M. K., and Buchhorn, M.: The spatial and temporal influence of infrastructure and road dust on seasonal snowmelt, vegetation productivity, and early season surface water cover in the Prudhoe Bay Oilfield, Arctic Science, pp. 1–26, https://doi.org/10.1139/as-2022-0013, 2022.

Brandt, J.: The extent of the North American boreal zone, Environmental Reviews, 17, 101–161, https://doi.org/10.1139/A09-004, 2009.

Burn, C. R. and Kokelj, S. V.: The environment and permafrost of the Mackenzie Delta area, Permafrost and Periglacial Processes, 20, 83–105, https://doi.org/10.1002/ppp.655, 2009.

Eaton, A. K., Rouse, W. R., Lafleur, P. M., Marsh, P., and Blanken, P. D.: Surface Energy Balance of the Western and Central Canadian Subarctic: Variations in the Energy Balance among Five Major Terrain Types, Journal of Climate, 14, 3692–3703, https://doi.org/10.1175/1520-0442(2001)014<3692:SEBOTW>2.0.CO;2, 2001.

Environment and Climate Change Canada: Historical data Trail Valley, Northwest Territories, URL: https://climate.weather.gc.ca/, accessed: 2023-08-12, 2021.

Fraser, R. H., Lantz, T. C., Olthof, I., Kokelj, S. V., and Sims, R. A.: Warming-Induced Shrub Expansion and Lichen Decline in the Western Canadian Arctic, Ecosystems, 17, 1151–1168, https://doi.org/10.1007/s10021-014-9783-3, 2014.

Hammar, J.: Drivers of permafrost degradation along the Inuvik to Tuktoyaktuk Highway (ITH), Master's thesis, University of Potsdam, 2022.

Muster, S., Heim, B., Abnizova, A., and Boike, J.: Water body distributions across scales: A remote sensing based comparison of three arctic tundrawetlands, Remote Sensing, 5, 1498–1523, https://doi.org/10.3390/rs5041498, 2013.

O'Neill, H. B., Smith, S. L., Burn, C. R., Duchesne, C., and Zhang, Y.: Widespread Permafrost Degradation and Thaw Subsidence in Northwest Canada, Journal of Geophysical Research: Earth Surface, 128, e2023JF007 262, https://doi.org/10.1029/2023JF007262, e2023JF007262 2023JF007262, 2023.

Palmer, M., Burn, C., Kokelj, S., and Allard, M.: Factors influencing permafrost temperatures across tree line in the uplands east of the Mackenzie Delta, 2004-2010, Canadian Journal of Earth Sciences, 49, 877–894, https://doi.org/10.1139/e2012-002, 2012.

Raynolds, M. K., Walker, D. A., Balser, A., Bay, C., Campbell, M., Cherosov, M. M., Daniëls, F. J., Eidesen, P. B., Ermokhina, K. A., Frost, G. V., Jedrzejek, B., Jorgenson, M. T., Kennedy, B. E., Kholod, S. S., Lavrinenko, I. A., Lavrinenko, O. V., Magnússon, B., Matveyeva, N. V., Metúsalemsson, S., Nilsen, L., Olthof, I., Pospelov, I. N., Pospelova, E. B., Pouliot, D., Razzhivin, V., Schaepman-Strub, G., Šibík, J., Telyatnikov, M. Y., and Troeva, E.: A raster version of the Circumpolar Arctic Vegetation Map (CAVM), Remote Sensing of Environment, 232, 111 297, https://doi.org/10.1016/j.rse.2019.111297, 2019.